# Tackling Structural Complexity in Li_2_S-P_2_S_5_ Solid-State Electrolytes Using Machine Learning Potentials

**DOI:** 10.3390/nano12172950

**Published:** 2022-08-26

**Authors:** Carsten G. Staacke, Tabea Huss, Johannes T. Margraf, Karsten Reuter, Christoph Scheurer

**Affiliations:** 1Fritz-Haber-Institut der Max-Planck-Gesellschaft, Faradayweg 4-6, 14195 Berlin, Germany; 2Forschungszentrum Jülich GmbH, Institute of Energy and Climate Research, Fundamental Electrochemistry (IEK-9), Wilhelm-Johnen-Straße, 52428 Jülich, Germany

**Keywords:** machine learning, amorphous, Li-ion battery, high ionic conductivity solid electrolyte

## Abstract

The lithium thiophosphate (LPS) material class provides promising candidates for solid-state electrolytes (SSEs) in lithium ion batteries due to high lithium ion conductivities, non-critical elements, and low material cost. LPS materials are characterized by complex thiophosphate microchemistry and structural disorder influencing the material performance. To overcome the length and time scale restrictions of *ab initio* calculations to industrially applicable LPS materials, we develop a near-universal machine-learning interatomic potential for the LPS material class. The trained Gaussian Approximation Potential (GAP) can likewise describe crystal and glassy materials and different P-S connectivities P${}_{m}$S${}_{n}$. We apply the GAP surrogate model to probe lithium ion conductivity and the influence of thiophosphate subunits on the latter. The materials studied are crystals (modifications of Li${}_{3}$PS${}_{4}$ and Li${}_{7}$P${}_{3}$S${}_{11}$), and glasses of the *x*Li${}_{2}$S–(100 – *x*)P${}_{2}$S${}_{5}$ type (*x* = 67, 70 and 75). The obtained material properties are well aligned with experimental findings and we underscore the role of anion dynamics on lithium ion conductivity in glassy LPS. The GAP surrogate approach allows for a variety of extensions and transferability to other SSEs.

## 1. Introduction

While lithium-ion batteries with liquid electrolytes entered the market in 1991, all-solid-state lithium-ion batteries (ASS-LIBs), although investigated for decades, are still not widely in use [1,2,3,4,5,6]. ASS-LIBs promise several advantages in comparison to liquid electrolyte batteries: higher power density, minimized safety and fire hazards, longer cycle lifetimes, more comprehensive temperature ranges, and enhanced energy density by potential usage of Li metal anodes [1,7,8]. Solid electrolytes of the Li${}_{2}$S-P${}_{2}$S${}_{5}$ material class have gained substantial attention due to their favorable properties [6,9]. First, they possess high conductivities of up to 10${}^{-2}$ Scm${}^{-1}$, which ranks them among the most conductive solid electrolytes such as Li${}_{10}$GeP${}_{2}$S${}_{12}$ or Li${}_{1.3}$Al${}_{0.3}$Ti${}_{1.7}$(PO${}_{4}{)}_{3}$ [10,11]. Secondly, they are composed of the earth-abundant elements sulfur and phosphorous enabling sustainable applications at large scales.

However, this material class’s potential is hampered by the poor understanding of the relevant structure-property relations. This manifests itself in huge deviations in Li-ion conductivity between theory and experiment. As such, β-Li${}_{3}$PS${}_{4}$ serves as an illustrating example. Experimental studies report a lithium ion conductivity of approximately $10^{-7}$ Scm${}^{-1}$, making the material unsuitable for industrial battery applications [12]. In contrast, an *ab initio* study predicts a conductivity of $10^{-1}$ Scm${}^{-1}$; a six orders of magnitude deviation from experiment that would make the material the new record holder in solid-state lithium ion conduction [13]. Such huge discrepancies often arise from computational limitations that constrain tractable system sizes and sampling times. In the LPS case, high-resolution TEM images for instance revealed the presence of crystalline nanoparticles in otherwise amorphous regions, highlighting that conductivity calculations of ideal crystals are too short-sighted for this materials class [14]. The problem is further accentuated by the complex chemical structure of LPS [15,16]. A large structural variability at the molecular level, more precisely different thiophosphate poly-anions, are found in all crystalline and amorphous materials [6,17]. For a detailed description of the lithium ion conductivity in LPS we thus need to tackle these challenges: structural complexity of LPS glass compounds, chemical reactivity of thiophosphates, and the influence of anion composition on the lithium ion conductivity.

Here we tackle these challenges by replacing the computationally demanding direct first-principles calculations with a surrogate machine-learning (ML) model. Once trained, this Gaussian Approximation Potential (GAP) model allows for an upscaling of both time and length scale: molecular dynamics (MD) simulations covering up to several nanoseconds and system sizes of several thousand atoms become feasible. Furthermore, the flexibility offered by the ML approach allows one to implement a GAP model that is more versatile and can better represent the crucial complex chemistry than a classical force field [18]. We present a data-efficient iterative training approach to extend an earlier ML force field to yield a near-universal description of the LPS material class [19,20].

In the first part of this work we present our data-efficient training protocol and evaluate the GAP model on (a) its predictive accuracy for lithium ion conductivity and (b) its ability to reproduce two known phase transitions in crystalline Li${}_{3}$PS${}_{4}$. The second part focuses on the influence of anion composition on the lithium ion conductivity of different LPS glass compounds.

## 2. Methods

### Computational Details

Reference density-functional theory (DFT) calculations are performed with the PBE functional, default ’light’ integration grids and a ’tier 1’ basis set of numerical atomic orbitals, as implemented in FHI-aims [21,22]. The Brillouin zone is sampled with a 1 × 1 × 1 k-grid. Initial training configurations are generated with *ab initio* molecular dynamics (MD) using the Γ-point approximation for the k-grid. GAP-based MD and Nudged-Elastic-Band (NEB) simulations are performed using the LAMMPS [23] code and the corresponding interface to QUIP [24,25]. For training set construction and data analysis, the atomic simulation environment ASE, SciPy and scikit-learn are used [26,27,28].

## 3. Results

### 3.1. Lithium Ion Mobility

We obtain the reactive GAP model used to describe the LPS class by fitting to DFT training data computed with the FHI-aims full-potential package [22]. The underlying approach is based on three consecutive steps: defining the anion lattice, sampling of Li-sites, and fine-tuning the materials density. In the first place, only the dominant anion species (e.g., PS${}_{4}^{3-}$ and P${}_{2}$S${}_{7}^{4-}$) are taken into consideration and utilized in a ratio that represents the desired stoichiometry correctly. For a data-efficient sampling of lithium sites, we sample Li-ion distributions on stable and meta-stable Li sites in a quasi-Monte Carlo like fashion. The materials density is obtained by an iterative compression scheme. Convergence, a detailed step-by-step description of the underlying algorithm, and numerical error assessments of the training procedure are given in Sections A–C in the Appendix A. The benefit of this approach is that it allows the free tuning of stoichiometries and polyanion ratios. In contrast to previous work on crystalline Li${}_{7}$P${}_{3}$S${}_{11}$ we use a purely short-ranged GAP. In Ref. [19] we combined a GAP model with an electrostatic baseline in order to study the role of long-range electrostatics in machine-learned interatomic potentials for complex battery materials. We previously showed that neglecting long-range electrostatics is unproblematic for describing lithium ion transport in isotropic bulk-like systems [19].

As a first validation of our GAP model, we turn to the Li-ion conductivity of crystalline LPS materials (α, β, γ Li${}_{3}$PS${}_{4}$ and Li${}_{7}$P${}_{3}$S${}_{11}$) at finite temperature, predicted from MD simulations via the Nernst-Einstein equation (see section F in the Appendix A for details). Using the GAP model we evaluate the ionic conductivity from the mean-square-displacement (MSD) sampled during 2 ns MD simulations for every crystalline compound at various temperatures between 400 and 800 K. Room temperature (RT) conductivities are extrapolated from a linear fit. Note that for crystalline Li${}_{7}$P${}_{3}$S${}_{11}$ we required longer simulation times of up to 13 ns to reach converged conductivities, i.e., time scales that would essentially be prohibitive for direct *ab initio* MD. While Li-ion conductivity in LPS is usually dominated by diffusion of defects (Li${}^{+}$ vacancies), Li${}_{7}$P${}_{3}$S${}_{11}$ exhibits a more collective Li${}^{+}$ motion yielding the observed high conductivity [19,29,30]. As seen in Figure 1, a broad range of Li-ion conductivities are exhibited in LPS.

While high RT conductivities of up to 3.6 × 10${}^{-3}$ Scm${}^{-1}$ are found for α-Li${}_{3}$PS${}_{4}$ and Li${}_{7}$P${}_{3}$S${}_{11}$, β and γ-Li${}_{3}$PS${}_{4}$ exhibit poor RT conductivities of 10${}^{-5}$ to 10${}^{-7}$ Scm${}^{-1}$. These crystalline RT conductivities are in good agreement with experimental literature, although the extrapolated RT conductivity of β-Li${}_{3}$PS${}_{4}$ is somewhat overestimated [30]. For ensemble averaging, we generated 20 structurally uncorrelated glass geometries for each specific temperature and stoichiometry. Hence, each data point in Figure 1 is an average over 20 MD calculations [31]. We consider three different stoichiometries in the analysis that cover the range from fully tetrahedral (Li${}_{3}$PS${}_{4}$) via mixed (Li${}_{7}$P${}_{3}$S${}_{11}$) to fully bridged tetrahedral (Li${}_{4}$P${}_{2}$S${}_{7}$) thiophosphate moieties. These three dominant anion subunits are depicted below. As apparent from Figure 1, the ion conductivity over the whole temperature range and the extrapolated RT conductivities increase with growing Li${}_{2}$S content of the glass material, almost tripling conductivity from Li${}_{4}$P${}_{2}$S${}_{7}$ (Li${}_{2}$S = 67 mol%) to Li${}_{3}$PS${}_{4}$ (Li${}_{2}$S = 75 mol%). Hence, for an increasing Li${}_{2}$S content an increase RT conductivity is observed. These findings are again in good agreement with experimental studies.

### 3.2. Li${}_{3}$PS${}_{4}$ Phase Transition

As a final validation step, we test the GAP’s predictive power on the known phase transitions in Li${}_{3}$PS${}_{4}$. As we show in Appendix A in the Appendix A, the Arrhenius curves of β and γ-Li${}_{3}$PS${}_{4}$ exhibit a change of slope at roughly 700 K. Above 700 K, conductivities of β and γ-Li${}_{3}$PS${}_{4}$ even match those of α-Li${}_{3}$PS${}_{4}$. This change of slope is caused by the phase transition to α-Li${}_{3}$PS${}_{4}$, involving a rotation of 25 % of the PS${}_{4}^{3-}$ tetrahedra by 180${}^{\circ }$ for both structures [18].

We can probe the phase transition quantitatively by studying the radial distribution functions (RDFs) of the sulfur sublattice as a function of simulation temperature (Figure 2). The β- and γ-phase share a HCP (hexagonal close-packed) sulfur sublattice, which is transformed to a BCC (body centered cubic) lattice in the α-phase [18]. For both sublattices, the S-S RDF displays a distinct peak at 3.4 Å, attributed to the intramolecular S-S distance. In the HCP sublattice, a second distinct peak at 4.3 Å is observed. The latter is missing in the BCC structure. Both β- and γ-phase show the characteristic double-peak in the low-temperature RDF, while the second peak vanishes for temperatures above 650 K. This same phase transition has also been observed in experimental studies and *ab initio* simulations [32,33]. Conceptually, these three phases can be distinguished by their different arrangement of PS${}_{4}^{3-}$. These are either all pointing in the same direction (γ), are arranged in a zig-zag fashion in one (α) or two directions (β) in space. A visualization can be found in Appendix A in the Appendix A [32]. The here obtained temperature between 600 and 700 K for the phase transition again matches fairly well with the experimentally reported 746 K [33].

### 3.3. The Role of Anion Composition in Li${}_{2}$S-P${}_{2}$S${}_{5}$ Glasses

Concluding that we can correctly describe the lithium ion dynamics and structural changes in crystalline Li${}_{3}$PS${}_{4}$ we now turn to the influence of anion composition on the lithium ion conductivity in LPS glasses. As shown above, the RT conductivity generally increases with Li${}_{2}$S content of the glass material. The increasing Li-ion conductivity is partly attributable to the lithium mass percentage increase at equal densities. This larger concentration of charge carriers yields higher conductivities for similar diffusion coefficients, accounting for an increase in conductivity of ∼30%. As this is much less than the above described rough tripling of the conductivity, we suspect the different anion compositions in the sulfur sub-lattice to be another, dominant factor.

Existing data on the origin of ion conductivity suppression by the anion lattice is quite ambiguous. For example, experimental studies report a strong conductivity suppression by P${}_{2}$S${}_{6}^{4-}$, attributed to meso-scale precipitation of the non-conducting Li${}_{2}$P${}_{2}$S${}_{6}$ phase [34,35]. On the contrary, density of state calculations report that P${}_{2}$S${}_{7}^{4-}$ should suppress ion conduction at the atomic scale [36]. The charge transfer along the covalent bond between the P and the bridging S lowers the positive partial charge of the P centers, which supposedly attracts Li${}^{+}$ ions to the P${}_{2}$S${}_{7}^{4-}$ anions more strongly than the other thiophosphate anions. These are just two illustrative examples discussed as possible origins of ion conductivity suppression by the anion lattice.

First, we analyse the anion composition at different temperatures for all three stoichiometries. Violin plots depicting the building block distributions at different temperatures within the structure ensembles are displayed in Figure 3. For the Li${}_{3}$PS${}_{4}$ glass, the simple PS${}_{4}^{3-}$ortho-thiophosphate is as intuitively expected the predominant species over the whole temperature range. Hypo-thiodiphosphate P${}_{2}$S${}_{6}^{4-}$ occurs only in small concentrations ≤10 at.% and shows no strong temperature dependence. Up to 25 at.% of pyro-thiodiphosphate P${}_{2}$S${}_{7}^{4-}$ occur at the lower temperature but gradually disappear between 600 and 700 K. These anion ratios are in agreement with experimental ratios found for Li${}_{3}$PS${}_{4}$ [37]. In both, the Li${}_{7}$P${}_{3}$S${}_{11}$ and Li${}_{4}$P${}_{2}$S${}_{7}$ glasses, the P${}_{2}$S${}_{6}^{4-}$ content instead increases between 400 and 700 K, even though the increase is not too pronounced in comparison to the width of the distribution in the ensemble. The found P${}_{2}$S${}_{6}^{4-}$ contents in Li${}_{7}$P${}_{3}$S${}_{11}$ and Li${}_{4}$P${}_{2}$S${}_{7}$ are slightly higher compared to experimental data [17].

Next, we analyze the number of Li-positions occupied during MD simulations at finite temperatures, by calculating the isosurface of the probability density distribution of Li-positions (exemplary visualizations see Appendix A in the Appendix A). When referencing the volume enclosed by the isosurface to the total volume of the simulation cell, we identify the relative accessible volume for all Li-ions for a given stoichiometry. As shown in the left panel of Figure 4 for the Li${}_{7}$P${}_{3}$S${}_{11}$ and Li${}_{4}$P${}_{2}$S${}_{7}$ glasses, the same relative volume is accessed by Li, while Li${}_{3}$PS${}_{4}$ exhibits a 10 % higher accessible Li-volume at all temperatures. This is intuitive as P${}_{2}$S${}_{n}^{4-}$ moieties have a larger surface/volume of the anion, allowing for a smaller number of Li-sites in the material compared to smaller PS${}_{4}^{3-}$ anions.

In order to explore the effect of the anion lattice motion on the Li-ion conductivity we either constrain the sulfur positions, or the phosphorous positions, and compare the Li ion conductivity obtained within MD simulations with these two frozen lattices to the unconstrained Li-ion conductivity. As seen in the right panel of Figure 4, the Li-ion conductivity decreases for all three glass stoichioemetries for both frozen lattices. However, while in the case of frozen phosphorus we observe only a slight decrease in conductivity, freezing the sulfur degrees of freedom reduces the conductivity by approximately two orders of magnitude. We observe the largest decrease of the conductivity for the stoichiometry consisting of the highest PS${}_{4}^{3-}$ (Li${}_{3}$PS${}_{4}$) content, and the smallest change for the lowest PS${}_{4}^{3-}$ content (Li${}_{4}$P${}_{2}$S${}_{7}$). This suggests that the motion of sulfur throughout the Li-ion conduction plays a significant role.

Smith and Siegel showed that in glassy Li${}_{3}$PS${}_{4}$, lithium migration occurs via a mechanism that combines a concerted motion of lithium ions with re-orientations of PS${}_{4}^{3-}$ anions [38]. This effect, known as the ’paddlewheel’ mechanism, can directly attribute the increasing Li-ion conductivity with increasing PS${}_{4}^{3-}$ content. So far, the paddlewheel effect has only been shown in Li${}_{3}$PS${}_{4}$, but our results confirm this effect occurs as long as PS${}_{4}^{3-}$ is present. Hence, the conductivity of Li${}_{4}$P${}_{2}$S${}_{6}$ and Li${}_{4}$P${}_{2}$S${}_{7}$ decrease as well when the sulfur lattice is frozen, but the effect is not as pronounced as in Li${}_{3}$PS${}_{4}$. As the Li-ion conductivity of all three stoichiometries is almost identical when the sulfur lattice is frozen, this actually suggests that the higher accessible volume of Li in Li${}_{3}$PS${}_{4}$ arises from re-orientations of PS${}_{4}^{3-}$ anions. Hence, both effects can not be decoupled, but rather the re-orientation of PS${}_{4}^{3-}$ anions generates new Li sites. Together with the increased overall Li content, this thus fully rationalizes why the Li-ion conductivity increases with higher Li${}_{2}$S content.

## 4. Conclusions

All of the herein described effects, collective Li-ion motion of crystalline Li${}_{7}$P${}_{3}$S${}_{11}$, phase transitions of crystalline Li${}_{3}$PS${}_{4}$, and the conductivity/anion-composition relation in glassy LPS, could not be studied before by a single interatomic potential, preventing the relative identification of trends and common origins. While not only this can now be achieved by our machine learning surrogate model, the general structure of the training protocol furthermore allows for a variety of extensions, including additional selection criteria [20,39], using an electrostatic baseline in the model [40], doping with transition metals, and modeling of solid/solid interfaces [41,42]. We correspondingly see much prospects in the use of ML potentials to further elucidate atomic scale processes in complex battery materials.

## Figures and Tables

**Figure 1 nanomaterials-12-02950-f001:**
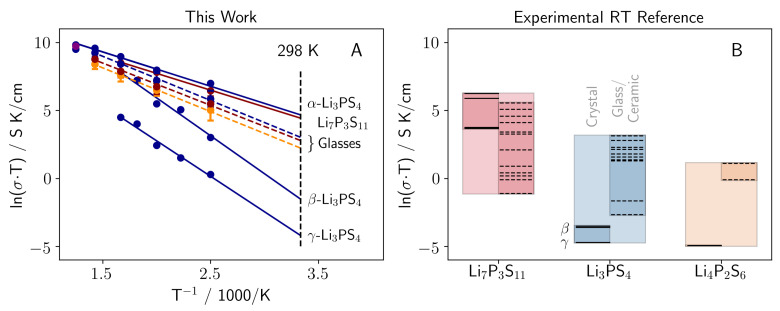
(**A**) Computational Arrhenius plots for Li${}_{7}$P${}_{3}$S${}_{11}$ (red solid line) and α, β and γ phase of Li${}_{3}$PS${}_{4}$ (blue solid lines), as well as the glasses of Li${}_{4}$P${}_{2}$S${}_{6}$ (orange dashed line), Li${}_{7}$P${}_{3}$S${}_{11}$ (red dashed line), and Li${}_{3}$PS${}_{4}$ (blue dashed line). (**B**) Reference conductivity data from literature. A tabulated form including references can be found in Appendix A in the Appendix A. Solid lines refer to nominal crystalline materials, dashed lines to glasses/ceramics.

**Figure 2 nanomaterials-12-02950-f002:**
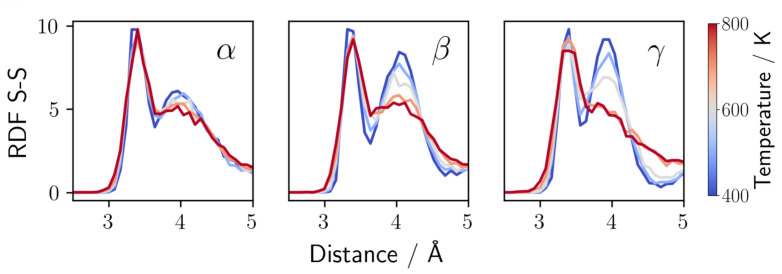
S-S radial distribution functions (RDFs) for MD snapshots of α-Li${}_{3}$PS${}_{4}$ (**left** panel), β-Li${}_{3}$PS${}_{4}$ (**middle** panel), and γ-Li${}_{3}$PS${}_{4}$ (**right** panel) at different temperatures. The disappearance of the peak at 4.3 Å, occurring for β and γ at 700K, corresponds to the phase transition to the α-phase.

**Figure 3 nanomaterials-12-02950-f003:**
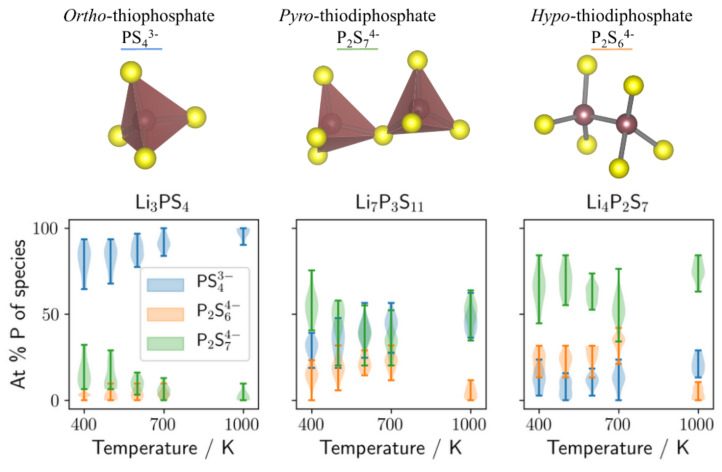
**Top**: Dominant anions in the Li${}_{2}$S–P${}_{2}$S${}_{5}$ material class. **Bottom**: Anion compositions for different MD temperatures, displayed for Li${}_{3}$PS${}_{4}$ (**left** panel), Li${}_{7}$P${}_{3}$S${}_{11}$ (**middle** panel), and Li${}_{4}$P${}_{2}$S${}_{7}$ (**right** panel) glasses.

**Figure 4 nanomaterials-12-02950-f004:**
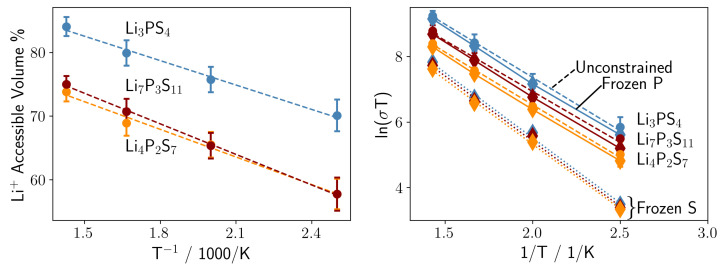
(**Left** panel) Accessible volume of lithium during MD simulations at various temperatures: Li${}_{3}$PS${}_{4}$ (blue dashed line), Li${}_{7}$P${}_{3}$S${}_{11}$ (red dashed line), and Li${}_{4}$P${}_{2}$S${}_{7}$ (orange dashed line). (**Right** panel) Li ion conductivity with frozen sulfur lattice (diamonds with dotted lines), with frozen phosphor lattice (diamonds with solid line) and without constraints on the sulfur lattice (dots with dashed line): Li${}_{3}$PS${}_{4}$ (blue lines), Li${}_{7}$P${}_{3}$S${}_{11}$ (red lines), and Li${}_{4}$P${}_{2}$S${}_{7}$ (orange lines).

## Data Availability

Data and codes can be found under Ref. [31].

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
