# Peer review of "Tackling Structural Complexity in Li2S-P2S5 Solid-State Electrolytes Using Machine Learning Potentials"

_nanomaterials, 2022, doi:10.3390/nano12172950_

Round 1

Reviewer 1 Report

I think the paper is well written.  If possible, the author describe more details in the following parts.

P2 l 82 "Note that we use a charge free approach in this work"

P3 l 101 "stoichiometry 20 glass structures"

P4 l123 " This same phase transition as also been observed in experimental studies and ab initio simulation"

Author Response

Response to Reviewer 1 Comments

We thank the reviewer for their positive feedback. We address all their recommendations in the following:

Reviewer: P2 l 82 "Note that we use a charge free approach in this work"

Reply: We have added the following paragraph to the discussion:

"In contrast to previous work on crystalline Li$_7$P$_3$S$_{11}$ we use a purely short-ranged GAP. In Ref [19] we combined a GAP model with an electrostatic baseline in order to study the role of long-range electrostatics in machine-learned interatomic potentials for complex battery materials."

Reviewer: P3 l 101 "stoichiometry 20 glass structures"

Reply: We have added the following paragraph to the discussion: 

"For ensemble averaging, we generated 20 structurally uncorrelated glass geometries for each specific temperature and stoichiometry. Hence, each data point in Fig. 1 is an average over 20 MD calculations."

Reviewer: P4 l123 " This same phase transition as also been observed in experimental studies and ab initio simulation"

Reply: We agree that a more substantial discussion of the three Li3PS4 phases is beneficial. We have therefore added the following description of these phases to the discussion and added Figure S6 to the SI:

"Conceptually, these three phases can be distinguished by their different arrangement of PS$_4^{3-}$. These are either all pointing in the same direction (gamma), are arranged in a zig-zag fashion in one (alpha) or two directions (beta) in space."

Reviewer 2 Report

The paper is well written an sound. It is well suited for a large audience and for people working in the field of lithium batteries. I recommend it for publication. I have only a minor concern: I feel that the right part of figure 1 is a little bit confusing because it is not clear which are the ranges where experimental data are available. Would it possible to present those data in a different way?

Author Response

Response to Reviewer 2 Comments

Reviewer: I have only a minor concern: I feel that the right part of figure 1 is a little bit confusing because it is not clear which are the ranges where experimental data are available. Would it possible to present those data in a different way?

Reply: We thank the reviewer for their positive feedback. We addressed the reviewers concern by splitting figure 1 into two panels and added the caption: A tabulated form including references can be found in table S3 in the SI.

Reviewer 3 Report

Manuscript named "Tackling structural complexity in Li2S-P2S5 Solid-State Electrolytes using Machine Learning” proposed a training protocol based on machine learning that can support various extensions, and studied the ion mobility, phase transition, and the relationship between anion and electrical conductivity in glassy lithium thiophosphate. For modern researches, machine learning as a method that can accelerate the experimental cycle and reduce the experimental cost deserves a lot of reports. The predicted results are in good agreement with the experimental phenomena. The article has some new ideas, needs to be considered on some minor issues, and can be published after a minor revision.

1. In Figure 1, the number of each picture should be correctly represented.

2. Noting that the Computational Details section is similar to the supporting Information. Is there any additional content in the supporting Information?

3. The differences in spatial structure of α, β and γ - Li3PS4 should be properly explained.

4. Supporting Information should be more numbered rather than just saying "see SI".

5. The explanation in line 105 to 108 is unclear as to how to reflect the relationship between Li2S content and RT conductivities.

Author Response

Response to Reviewer 2 Comments

Point 1.: In Figure 1, the number of each picture should be correctly represented.

Reply:  We thank the reviewer for their positive feedback. We addressed the concern by splitting figure 1 into two panels and labeled them accordingly.

Point 2: Noting that the Computational Details section is similar to the supporting Information. Is there any additional content in the supporting Information?

Reply:  We thank the reviewer. We cleared all redundancies of the SI and the main script.

Point3:  The differences in spatial structure of Li3PS4 should be properly explained.

Reply:  We agree that a more substantial discussion of the three Li3PS4 phases is benefical. We have therefore added the following description of these phases to the discussion and added Figure S6 to the SI:

"Conceptually, these three phases can be distinguished by their different arrangement of PS43-. These are either all pointing in the same direction (gamma), are arranged in a zig-zag fashion in one (alpha) or two directions (beta) in space."

Point 4: Supporting Information should be more numbered rather than just saying "see SI".

Reply:  We added the numbering of all figures in the  SI to the main script.

Point 5: The explanation in line 105 to 108 is unclear as to how to reflect the relationship between Li2S content and RT conductivities. 

\bigskip

Reply: We added the following paragraph to the discussion:

"As apparent from Figure 1, the ion conductivity over the whole temperature range and the extrapolated RT conductivities increase with growing Li2S content of the glass material, almost tripling conductivity from Li4P2S7 (Li2S = 67 mol%) to Li3PS4 (Li$_2$S = 75 mol%). Hence, for an increasing Li2S content, an increase RT conductivity is observed."